# A machine learning approach in a monocentric cohort for predicting primary refractory disease in Diffuse Large B-cell lymphoma patients

**Marie Y. Detrait**[1]*, **Stéphanie Warnon**[2], **Raphaël Lagasse**[1,3,4], **Laurent Dumont**[1], **Stéphanie De Prophétis**[5], **Amandine Hansenne**[5], **Juliette Raedemaeker**[5], **Valérie Robin**[5], **Géraldine Verstraete**[5], **Aline Gillain**[2], **Nicolas Depasse**[1], **Pierre Jacmin**[1], **Delphine Pranger**[5]

1 Department of Technology and Information Systems, Grand Hôpital de Charleroi, Charleroi, Belgium, 2 Department of Clinical Research, Grand Hôpital de Charleroi, Charleroi, Belgium, 3 Department of Medico-Economic Information, Grand Hôpital de Charleroi, Charleroi, Belgium, 4 School of Public Health, Université Libre de Bruxelles (U.L.B.), Brussels, Belgium, 5 Division of Hematology, Hematology and oncology Department, Grand Hôpital de Charleroi, Charleroi, Belgium

* marie.detrait@ghdc.be

## Abstract

### Introduction

Primary refractory disease affects 30–40% of patients diagnosed with DLBCL and is a significant challenge in disease management due to its poor prognosis. Predicting refractory status could greatly inform treatment strategies, enabling early intervention. Various options are now available based on patient and disease characteristics. Supervised machine-learning techniques, which can predict outcomes in a medical context, appear highly suitable for this purpose.

### Design

Retrospective monocentric cohort study.

### Patient population

Adult patients with a first diagnosis of DLBCL admitted to the hematology unit from 2017 to 2022.

### Aim

We evaluated in our Center five supervised machine-learning (ML) models as a tool for the prediction of primary refractory DLBCL.

### Main results

One hundred and thirty patients with Diffuse Large B-cell lymphoma (DLBCL) were included in this study between January 2017 and December 2022. The variables used for analysis included demographic characteristics, clinical condition, disease characteristics, first-line

**Data Availability Statement:** All files are available at https://github.com/MarieDetrait-MD/PrimaryRefractoryDLBCL.

**Funding:** The author(s) received no specific funding for this work.

**Competing interests:** The authors have declared that no competing interests exist.

therapy and PET-CT scan realization after 2 cycles of treatment. We compared five supervised ML models: support vector machine (SVM), Random Forest Classifier (RFC), Logistic Regression (LR), Naïve Bayes (NB) Categorical classifier and eXtreme Gradient Boost (XGboost), to predict primary refractory disease. The performance of these models was evaluated using the area under the receiver operating characteristic curve (ROC-AUC), accuracy, false positive rate, sensitivity, and F1-score to identify the best model. After a median follow-up of 19.5 months, the overall survival rate was 60% in the cohort. The Overall Survival at 3 years was 58.5% (95%CI, 51–68.5) and the 3-years Progression Free Survival was 63% (95%CI, 54–71) using Kaplan-Meier method. Of the 124 patients who received a first line treatment, primary refractory disease occurred in 42 patients (33.8%) and 2 patients (1.6%) experienced relapse within 6 months. The univariate analysis on refractory disease status shows age ($p = 0.009$), Ann Arbor stage ($p = 0.013$), CMV infection ($p = 0.012$), comorbidity ($p = 0.019$), IPI score ($p<0.001$), first line of treatment ($p<0.001$), EBV infection ($p = 0.008$) and socio-economics status ($p = 0.02$) as influencing factors. The NB Categorical classifier emerged as the top-performing model, boasting a ROC-AUC of 0.81 (95% CI, 0.64–0.96), an accuracy of 83%, a F1-score of 0.82, and a low false positive rate at 10% on the validation set. The eXtreme Gradient Boost (XGboost) model and the Random Forest Classifier (RFC) followed with a ROC-AUC of 0.74 (95%CI, 0.52–0.93) and 0.67 (95%CI, 0.46–0.88) respectively, an accuracy of 78% and 72% respectively, a F1-score of 0.75 and 0.67 respectively, and a false positive rate of 10% for both. The other two models performed worse with ROC-AUC of 0.65 (95%CI, 0.40–0.87) and 0.45 (95%CI, 0.29–0.64) for SVM and LR respectively, an accuracy of 67% and 50% respectively, a f1-score of 0.64 and 0.43 respectively, and a false positive rate of 28% and 37% respectively.

## Conclusion

Machine learning algorithms, particularly the NB Categorical classifier, have the potential to improve the prediction of primary refractory disease in DLBCL patients, thereby providing a novel decision-making tool for managing this condition. To validate these results on a broader scale, multicenter studies are needed to confirm the results in larger cohorts.

## Introduction

Diffuse large B-cell lymphoma (DLBCL) is the most common type of non-Hodgkin lymphoma and is currently curable for many patients with frontline immuno-chemotherapy [1]. However, around 30–40% of patients are unresponsive or experience early relapse [2, 3]. The prognosis for primary refractory patient is poor, and managing and treating these cases is a significant challenge due to disease heterogeneity and a complex genetic framework. The reasons for refractoriness are varied and include genetic abnormalities, alterations in the tumor and tumor microenvironment, and patient-related factors such as comorbidities, which can also influence treatment outcomes [1–4]. For progressive disease, an online tool based on the Cox Model is available to estimate patient survival [5].

Treatment options for refractory disease may include salvage chemotherapy followed by autologous hematopoietic stem cell transplantation (HSCT) for eligible patients. Another approach is the use of chimeric Ag receptor (CAR) T-cell therapy or inclusion in clinical trial

with targeted therapies, sometimes involving combinations of various regimens depending on patient eligibility [2].

Recently, advances in Machine learning (ML) have demonstrated its usefulness in analyzing large and complex datasets. In medicine and oncology, machine learning is valuable for creating predictive, diagnostic, or prognostic tools based on data-driven analytic approaches and the integration of various risk factors and parameters [6, 7]. As a subdomain of artificial intelligence (AI), ML has the ability to autonomously uncover patterns within datasets. It offers algorithms that can learn from examples to perform a task automatically. Artificial intelligence and ML are also transforming oncology by enhancing risk assessment, early diagnosis, prognosis estimation, and treatment selection. These technologies have shown greater accuracy than clinicians in predicting various cancer, including breast, brain, lung, liver, and prostate cancers [8]. Machine learning algorithms improve diagnosis and prognosis, leading to better patient outcomes and quality of life. Ongoing development and improvement of these technologies are essential for maximizing their benefits in cancer care, understanding the limitations and working on future prospects [8, 9].

Machine learning is closely related to mathematical models and statistical methods. Three types of learning can be computed: supervised, unsupervised and reinforcement learning. Supervised learning is particularly useful type for our purposes because its aim is outcome prediction. In supervised learning, the algorithm is provided with training data that contains both the recorded observations and the corresponding labels for the categories of those observations. The model is trained on a dataset containing information about patients, diseases, and associated outcomes for each individual. When fed with new observations, the model estimates the predicted likelihood of the output label based on the same features used in the training process [6–8]. In hematology, a review article by Shouval et al. [10] described previous studies that used machine learning for outcome prediction and diagnosis based on images or laboratory data. The current applications in laboratory hematology are described in a review article by Obstfeld et al. [11] with innovative uses in various subdomains, emphasizing how these technologies enhance standardization and efficiency, freeing up resources for more critical tasks such as patient care and research.

Recent studies highlight the effectiveness of machine learning in advancing the understanding and treatment of DLBCL. One study used EcoTyper, a machine-learning framework integrating transcriptome deconvolution and single-cell RNA sequencing, to characterize clinically relevant DLBCL cell states and ecosystems. This approach identified five malignant B-cell states and variations in 12 other cell lineages within the tumor microenvironment, revealing significant clinical heterogeneity and new therapeutic targets [12]. Another study the molecular mechanisms of matrine in treating DLBCL. Using various machine learning algorithms, researchers identified 222 matrine target genes and 1269 DLBCL hub genes, with five core target genes (CTSL, NR1H2, PDPK1, MDM2, and JAK3) showing stable binding and strong associations with DLBCL. Functional analysis suggested matrine's therapeutic effects could involve the PI3K-Akt signaling pathway [13]. These studies demonstrate the successful application of machine learning in uncovering new insights and potential treatments for DLBCL.

To develop a model for predicting the risk of primary refractory DLBCL, we tested five machine-learning algorithms using parameters obtained from a monocentric dataset. Based on recent literature in Machine Learning and DLBCL, we believe our test is feasible in this setting due to the availability of comprehensive and relevant clinical data, which enables robust model training and validation. We expected that our results could provide a clinically useful model for refractory disease prediction and demonstrate the machine learning usefulness in the DLBCL management.

## Methods

### Study population

The Grand Hôpital de Charleroi is a large non-academic medical Center with 1154 beds and one of the most important oncology and hematology departments in the Walloon Region of Belgium. For this retrospective study, we identified a retrospective cohort of consecutive adult patients (age ≥18 years old) with a first diagnosis of DLBCL treated in the Hematology department at Grand Hôpital de Charleroi between 1 January 2017 and 31 December 2022. Patients who chose to be treated at another hospital were excluded. Patient data were extracted from our oncological-hematological database (RegistreOncoHematoGHdC, FilemakerPro v.17) to construct the dataset.

**Ethics.**   The patient records were anonymized upon extraction and prior to analysis. The authors did not have access to any information that could identify the patients at any time. The study was conducted in accordance with the Declaration of Helsinki. Our protocol complied with Belgian regulatory requirements and was reviewed and approved by the Ethics Committee of Grand hôpital de Charleroi (G2-2023-E006).

**Data analysis and statistics.**   The features set was curated with relevant variables tailored to our objectives and based on structured data from our database (RegistreOncoHematoGHdC). Data was updated and extracted on June 30, 2023. For variable description, categorical variables were presented as numbers and percentages, while discrete/continuous variables were described using median values along with the range (min-max). Comparisons between features (predictors) and target (primary refractory state) were performed using chi-squared tests for categorical variables and the Kruskal-Wallis test for discrete/continuous variables.

This analysis was complemented by a Decision Tree Classifier (DTC) to identify the most influential features as determined by their feature importance. Feature importance refers to a measure of the relative importance of different variables used by the model to make predictions. In a DTC, feature importance is assessed by measuring the impact of each variable on reducing the impurity of the model's nodes. The impurity of a node represents the diversity of class labels within the node, and common criteria for measuring impurity include the Gini index and entropy.

Overall survival and Progression Free Survival (PFS) were calculated using the Kaplan-Meier method [14]. The following variables were extracted for analysis: age, body mass index (BMI), first department or general practitioner that refers patient, time intervals between the first department and the hematology department, time elapsed between diagnosis and treatment, first line of treatment, second line of treatment, third line of treatment, CAR-T cell therapy, autologous HSCT, date of diagnosis, SARS-COV-2-19 pneumonia, positron emission tomography/computed tomography (PET/CT) scan performed after 2 cycles of treatment, overweight, malnutrition, disease status at the end of the study, relapse, date of relapse, time between diagnosis and relapse, response at first line of treatment, living status at the end of the study, follow-up time, survival time, cause of death, date of last contact, patient lost, Germinal Center (GC) phenotype, MYC mutation, extra copies of cMYC, BCL6 mutation, BCL2 mutation, 'double HIT', 'triple HIT', International Prognosis Index (IPI) score, Ann Arbor stage, smoking, Alcohol consumption, Human Immunodeficiency Virus (HIV) seropositive, cytomegalovirus (CMV) infection, Epstein Barr virus (EBV) infection, Helicobacter pylori infection, socio-economics status, nervous system involvement, cerebral lymphoma, comorbidity and type of organ comorbidity. Using cytogenetic FISH studies, DLBCL with MYC and BCL2 and/or BCL6 rearrangements, are referred ad to as 'double hit' (DHL) or 'triple hit' (THL) lymphomas.

Statistical analysis was performed using Python (v3.9.13) and its library: Scipy.stats [15], Pingouin [16], Seaborn [17] and Lifelines [18]. The level of statistical significance was set at 0.05, with $p < 0.05$ considered statistically significant. We worked in the Anaconda environment (v 2.4.2) with Jupyter Notebooks (v6.5.4) [19].

**Model building.** For the purpose of predicting the probability of primary refractory disease, after selecting and removing unnecessary variables using data analysis, the following variables (predictors) were included in the dataset for models training and validation: gender, age, age category one (18–40), age category two (40–60), age category three (60–80), age category four ($> 80$), BMI, BMI category one (30–35), BMI category two (35–40), BMI category three ($>40$), first department or general practitioner that refers patient, time intervals between the first department and the hematology department, time elapsed between diagnosis and treatment, first line of treatment, SARS-COV-2-19 pneumonia, positron emission tomography/computed tomography (PET/CT) scan performed after 2 cycles of treatment, overweight, malnutrition, GC phenotype, 'double HIT', 'triple HIT', IPI score, Ann Arbor stage, smoking, Alcohol consumption, cytomegalovirus (CMV) infection, Epstein Barr virus (EBV) infection, socio-economics status, nervous system involvement, comorbidity.

There was no major class imbalance, with 42 (33.8%) observations of primary refractory disease status. The target was the response status after the first line of treatment. Within the cohort, 120 patients underwent a first line of treatment and were evaluated. To identify the best-performing algorithm for predicting the probability of primary refractory disease, we trained and optimized five different models based on previous literature: Support Vector Machine (SVM), Random Forest Classifier (RFC), Naïve Bayes (NB) Categorical Classifier, Logistic Regression (LR) and eXtreme Gradient Boost (XGBoost) [8–10].

In summary, SVM generalizes a linear classifier by finding the plane or hyperplane that offers the greatest separation between two classes. Random Forest generates a large number of decision trees based on random subsamples of the training set and returns the mode of the individual trees' predictions for classification tasks [20]. Logistic regression is a generalized form of linear regression that uses a logistic function to model the probability of a binary outcome from numerous covariates. Naïve Bayes applies Bayes' theorem with strong independence assumptions between the features. Naïve Bayes Categorical Classifier requires a small amount of training data to estimate the parameters necessary for classification and is particularly effective for categorical data [15]. XGBoost employs a more regularized form of gradient boosting with high performance. This algorithm is based on decision trees and uses sequential ensemble methods, also known as "boosting," to iteratively correct the mistakes of the previous models in the sequence [21].

Data preparation prior to algorithm processing involved preparing categorical variables for Random Forest using a One-hot Encoder, which is a type of binary format transformation. In this method, each category of a categorical variable is transformed into a new binary column. This transformation helps Random forest interpret and process the categorical data more effectively. StandardScaler was applied for SVM and Logistic Regression, this is particularly important for these algorithms that require normalized data to perform effectively. By standardizing the features, StandardScaler ensures that each feature contributes equally to the result, improving the performance and accuracy of these algorithms. Additionally, an OrdinalEncoder was used for categorical variables before applying XGBoost, NBC Classifier, SVM, and LR, which require numerical input. By using, OrdinalEncoder, the categorical variables are transformed into numerical values by assigning an integer to each category while preserving the ordinal relationship between the categories. The treatment of missing data involved applying 'SimpleImputer' and 'KNNImputer' before working with the RF model. The treatment of missing categorical data involved applying 'SimpleImputer', this method

fills in missing data with the most frequent value from the column. The missing discrete/continuous variables were treated with 'KNNImputer', this method is an advanced method that fills in missing values by looking at the nearest neighbors. It uses the values of similar data points (neighbors) to estimate and replace the missing values. This approach can provide accurate imputations by considering the relationships between different data points. Missing data were handled as follows for the other algorithms: on missing value for the extension at the nervous system was set to 'no' as the majority of observations. Additionally, there were 11 missing values for the delay between services, which we filled with the median (17.5 days). One missing value for the delay between diagnosis and treatment was filled with the median (6 days).

In our experiment, we set the random state to 1 ton ensure reproducibility. By fixing the random state, we guarantee that the operations involving randomness, such as data splitting and model initialization, produce the same results every time the experiment is running. This allows for consistent comparisons between different models and techniques, ensuring that our finding are reliable and can accurately replicated.

Since the dataset is limited, we have uniformly reduced the size of the validation set to 15% for all models, including XGBoost, NBC classifier, SVM (NuSVC), LR, and RF. A validation set with data not seen by the algorithms will be used later in a prospective study, which should lead to production deployment.

For each algorithm, a prediction model was trained and tested using 10-fold-cross-validation on 85% training sample to ensure models were each trained uniformly. A grid search approach was performed for RFC, SVM and XGBoost to obtain the best parameters for each model. The 15% holdout set was the validation set and was used to determine the model's performance characteristics with area under the curve from receiver operator characteristic (ROC-AUC), accuracy (true positive + true negative/total), false positive rate (false positive rate = 1- specificity), sensitivity (sensitivity = true positive/true positive + false negative), and F1-score (the harmonic mean of precision and recall). A confusion matrix was created to calculate the indicators of each model.

In development and evaluation of ML models, ROC-AUC evaluates the capacity of the classifier to discriminate between two classes and is one of the best metrics because this metric has the capability to encapsulate the effectiveness of a classifier in a single measurement. The optimal models are identified by achieving the largest Area Under the Curve (AUC) [22], making this metric the focal point of our analysis [23]. A model's discriminative ability is measured by ROC-AUC in selecting true positives and negatives while minimizing false positives. A model with an AUC of 0.90 to 0.99 is considered as excellent, 0.80 to 0.89 is good, 0.70 to 0.79 is fair and 0.51 to 0.69 is poor. The F1-score is also a valuable metric for evaluating the performance of a classification model. It is the harmonic mean of precision and recall, providing a single score that balances the trade-off between the two scores. Precision measures the accuracy of positive predictions (i.e., how many predicted positives are true positives), while recall measures the ability of the model to identify all relevant instances (i.e., how many actual positives are correctly identified). A model with a F1-score of 0.9–1.0 is considered as excellent, the model is highly accurate in predicting both positive and negative classes, 0.8–0.9 is a good performance, 0.7–08 is a fair performance, 0.6–0.7 is a poor performance and below 0.6 it is an unacceptable performance [24].

We worked in the Anaconda environment (v 2.4.2) with Jupyter Notebooks (v6.5.4) [19] and Scikit-learn (v1.2.2) to build and evaluate the models [25]. The dataset, the code and the preparation of data are available on https://github.com/MarieDetrait-MD/PrimaryRefractoryDLBCL.

## Results

We analyzed data from 13O patients aged from 25 to 95 years who were diagnosed with first-time DLBCL between January 2017 and December 2022. Data extraction and analysis were performed on 30 June 2023. Patient and disease characteristics are shown in Table 1. The median follow-up of the study was 19.5 months [range, 1–77] from diagnosis to death or the latest follow-up date. At the end of the study, the average age of the cohort was 66.84 years, with 72.3% of patients being over 60 years old. There were 69 male patients (53%) and 61 female patients (47%). Ninety-eight patients (75.3%) were in advanced stages (Ann Arbor stage III-IV) at diagnosis, and 43 patients (33%) had a high-risk IPI score. According to the Hans Criteria, 44 patients (33.85%) were classified into GCB subtype, and 86 (66.15%) into non-GCB subtype. Seventeen patients (13%) had double-HIT rearrangements (MYC and BCL2 or BCL6) and 3 patients (2.3%) had triple-HIT rearrangements (MYC and BCL2 and BCL6). The median time to refer a patient to the Hematology department was 17.5 days [range, 1–100 days] and the median time between diagnosis and treatment was 6 days [range, 1–58 days]. Sixty-eight patients (52.3%) had comorbidities; among them, 38 patients (29%) were either active or former smokers, and 14 patients (10%) reported excessive alcohol intake at the time of diagnosis.

Overall survival (OS) at 3 years was 58.5% (95%CI, 51–68.5) and the 3-years Progression Free Survival (PFS) was 63% (95%CI, 54–71) using Kaplan-Meier method. We also found 51% (95% CI, 38–61) for 5-years OS (Fig 1) and 57% (95% CI, 46–67.5) for 5-years PFS (Fig 2).

The first line of treatment for most patients was based on R-CHOP (R-mini-CHOP or R-CEOP). Younger patients received R-ACBVP, and R-MPVA was administered for cerebral lymphomas. Treatment characteristics and outcome are presented in Table 2. One hundred and twenty-four patients (95.3%) received a first line of treatment. Primary refractory disease occurred in 42 patients (33.8%) and two patients (1.6%) experienced relapse within 6 months. The median time to relapse was 10 months [range, 4–36]. At the end of the study, 78 patients (60%) were alive (on 130 patients) and among these, 70 (89%) were in complete remission. Causes of death were as follows: 27 patients (52%) from DLBCL, 9 patients (17%) from septic shock, 4 patients (7%) from SARS-CoV-2 pneumonia, 4 patients (7%) from cardiologic reasons, and another 8 patients due to various other causes, including other cancers. Among the 42 patients (33.8%) with primary refractory disease, 8 patients died from DLBCL before salvage therapy, 34 patients (77.3%) underwent salvage chemotherapy, and at the end of the study, 14 (33%) were alive and 10 (23.8%) were in remission.

The assessment of disease response during treatment using an interim PET/CT scan has been gradually implemented in most centers since 2017 [26]. In our center, the systematic use of interim PET/CT scans began gradually in 2018 and has been fully implemented since 2021. Therefore, in this cohort, only 38 patients (30%) had a PET/CT scan evaluation after 2 cycles of first line treatment. Among this sub-cohort, 16 patients (42%) were in complete response, 12 patients (31.5%) had a partial metabolic response, and 10 patients (26.3%) were refractory.

In univariate analysis, primary refractory disease was associated with patient age (p = 0.009), Ann Arbor stage (p = 0.013), CMV infection (p = 0.012), presence of comorbidity (p = 0.019), IPI score (p<0.001), first line of treatment (p<0.001), EBV infection (p = 0.008) and patient socio-economics status (p = 0.02). Interestingly, the most important features for a Decision Tree Classifier (DTC), based on feature importance (the contribution of each feature to the model's performance) were the IPI score (Feature-score = 54), BMI (Feature-score = 22), patient age (Feature-score = 9.3), time between diagnosis and treatment (Feature-score = 9.2), and the BMI category (35–40) (Feature-score = 4.4). The importance of the IPI score and patient age as predictive variables emerged from our data analysis.

**Table 1. Patient and disease characteristics.**

| N = 130 | % (n) |
|---|---|
| Gender | |
| Male | 53% (69) |
| Female | 47% (61) |
| Age, years | |
| Median (min-max) | 69.5 [25–95] |
| Age group, years | |
| 18–60 | 27.7% (36) |
| >60 | 72.3% (94) |
| Body Mass Index | |
| Median (min-max) | 21.71 [16.85–49.86] |
| Body Mass Index | |
| <17 | 0.7% (1) |
| 17–25 | 37.7% (49) |
| 25–30 | 37% (48) |
| 30–35 | 16.1% (21) |
| 35–40 | 3.8% (5) |
| >40 | 4.6% (6) |
| Time to refer a patient, days | |
| Median (min-max) | 17.5 [1–100] |
| Time between diagnosis and treatment, days | |
| Median (min-max) | 6 [1–58] |
| Ann Arbor stage | |
| I | 10% (13) |
| II | 13.8% (18) |
| III | 13% (17) |
| IV | 62.3% (81) |
| Missing | 0.7% (1) |
| International Prognosis Index | |
| Low risk [0–1] | 32.3% (42) |
| Low_intermediate risk [2] | 13.8% (18) |
| High intermediate risk [3] | 20.7% (27) |
| High risk [4–5] | 33% (43) |
| Subtype | |
| Germinal Center | 33.85% (44) |
| NON-Germinal Center | 66.15% (86) |
| MYC mutation | |
| Yes | 13.8% (18) |
| cMYC extracopy | |
| Yes | 10% (13) |
| BCL6 mutation | |
| Yes | 25.38% (33) |
| BCL2 mutation | |
| Yes | 17.7% (23) |
| Double HIT (rearrangements) | |
| MYC + BCL2 | 7.7% (10) |
| MYC + BCL6 | 5.3% (7) |
| Triple HIT (rearrangements) | |

(*Continued*)

**Table 1.** (Continued)

| N = 130 | % (n) |
| --- | --- |
| MYC + BCL2 + BCL6 | 2.3% (3) |
| Experienced SARS-COV-2-2019 infection | |
| Yes | 12.3% (16) |
| Helicobacter pylori infection at diagnosis | |
| Yes | 5.3% (7) |
| Epstein Barr Virus infection at diagnosis | |
| Yes | 3% (4) |
| Cytomegalovirus infection at diagnosis | |
| Yes | 1.5% (2) |
| Seropositive for Human Immunodeficiency Virus | |
| Yes | 0.7% (1) |
| Central Nervous System lymphoma | |
| Yes | 3% (4) |
| Socio-economics status | |
| No problem | 94.6% (123) |
| Precarity | 4.6% (6) |
| Homeless | 0.7% (1) |
| Excess alcohol intake | |
| Yes | 10.7% (14) |
| Active smoking or former smoking | |
| Yes | 29.2% (38) |
| Comorbidity | |
| Yes | 52.3% (68) |
| Other cancer | 0.7% (1) |
| Cardio-pulmonary disease | 9.2% (12) |
| Cardio-pulmonary-kidney disease | 42.30% (55) |
| Initial Medical Management | |
| General practitioner | 21.53% (28) |
| Hepato-Gastro-Enterology unit | 13% (17) |
| Oncology unit | 6.92% (9) |
| Pneumology unit | 5.38% (7) |
| Others (5 or less patients) | <5% (< or = 5 for 69 patients) |

## Machine learning models and performance

In order to identify the algorithm that provides the best classification results for predicting the probability of primary refractory disease, we evaluated the performances of each using ten-fold cross validation. The comparison of these performances is presented in Table 3.

The results showed that of the five ML algorithms, the NB Categorical classifier emerged as the best performing model, boasting a ROC-AUC of 0.81 (95% CI, 0.64–0.96), an accuracy of 83%, a F1-score of 0.82, and a low false positive rate at 10%. Following were the XGBoost model and the Random forest classifier with more modest results, ROC-AUC values of 0.74 (95% CI, 0.52–0.93) and 0.67 (95% CI, 0.46–0.88) respectively, and corresponding accuracy of 78% and 72% respectively, and false positive rate of 10% for both. The F1-score was 0.75 for XGBoost and 0.67 for RF. In contrast, SVM and LR performed less favorably with ROC-AUC values of 0.65 (95%CI, 0.40–0.87) and 0.45 (95%CI, 0.29–0.44) respectively and a high rate of

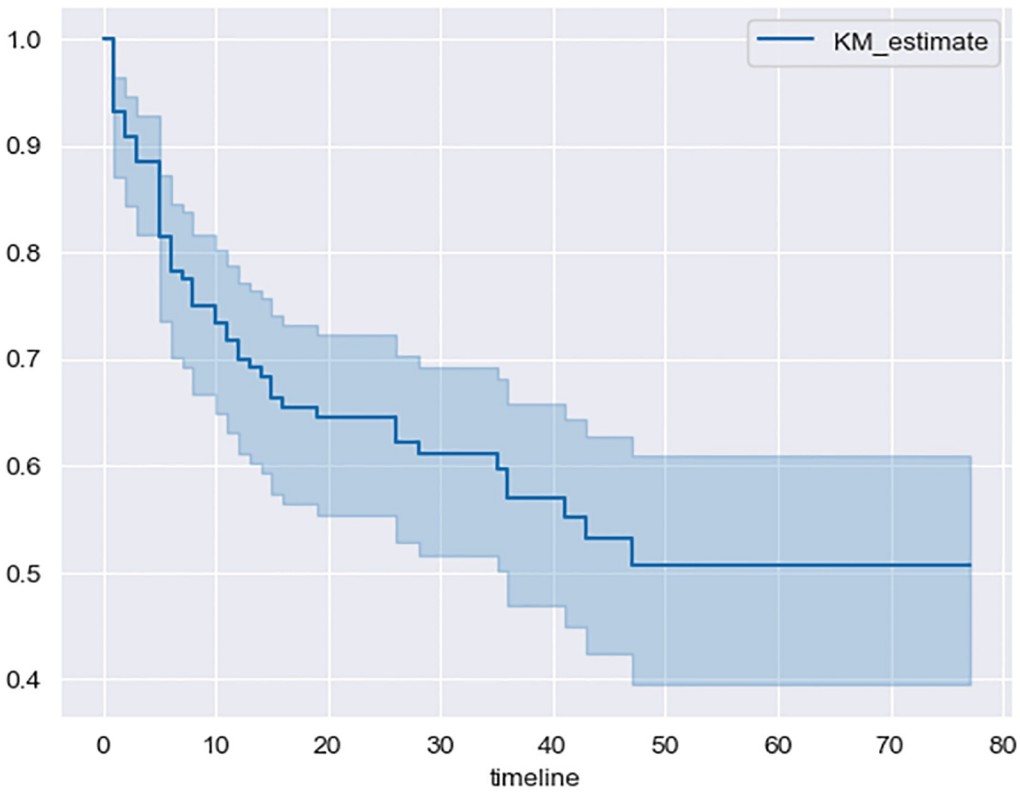

**Fig 1. Overall survival using Kaplan-Meier method.**

false positive with 28% of SVM and 37% for LR. The ROC-AUC curves for each model are shown in Fig 3.

The strengths and weaknesses are presented in Table 4. The differences of the NB Categorical classifier were notable, especially in terms of ROC-AUC and F1-score, indicating that it was able to better separate between both classes. In light of these results, we can observe that the NB Categorical classifier was the best alternative for building a model in order to predict primary refractory disease in DLBCL patients.

## Key features for the three best models

Based on feature importance, the top five features for the XGBoost model were the IPI score (Feature-score = 125), age category four ($> 80$) (Feature-score = 73), the BMI category (30–35) (Feature-score = 67), time elapsed between diagnosis and treatment (Feature-score = 69), and PET/CT scan performed after 2 cycles of treatment (Feature-score = 64). For the RFC, the top five features were age (Feature-score = 139), time between diagnosis and treatment (Feature-score = 102), the IPI score (Feature-score = 87), Ann Arbor stage (Feature-score = 86), and BMI (Feature-score = 79).

We also examined the log-likelihoods of conditional probabilities for various classes and variables using the 'feature_log_prob_' attribute in the Naive Bayes classifier. This attribute represents the log probabilities of each feature given each class in the Naïve Bayes classifier, helping to determine the contribution of individual features to the classification decision. The variables that contributed most to the model, in order of importance, were: 'triple hit', nervous

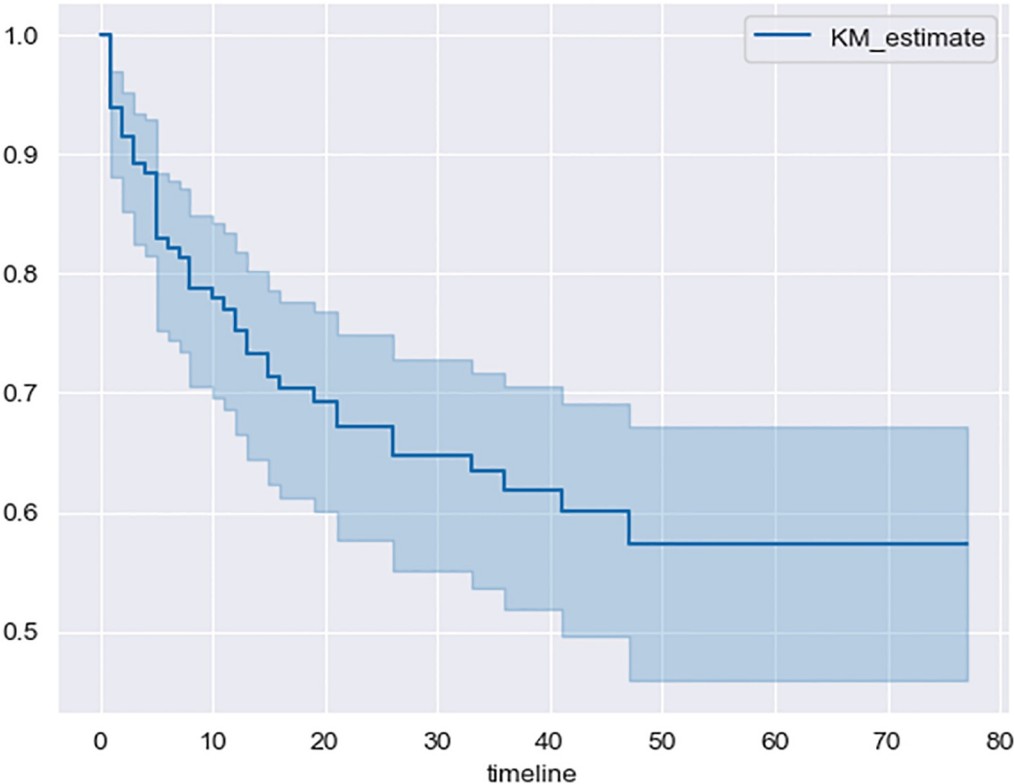

**Fig 2. Progression free survival using Kaplan-Meier method.**

system involvement, CMV infection, all age categories, all BMI categories, alcohol consumption, 'double hit', Ann Arbor stage, first line of treatment, first department or general practitioner that referred the patient, Germinal Center phenotype, smoking, comorbidity, gender, IPI score, SARS-COV-2-10 pneumonia, and overweight. This analysis identified key variables for the model.

## Discussion

As in several previous study in hematology, we found that using machine-learning techniques to evaluate outcomes in malignant diseases is beneficial. Recently, a model based on data mining and the alternating decision tree (ADT) demonstrated good performance in predicting transplantation mortality 100 days after allogeneic hematopoietic stem cell transplantation [27]. Another noteworthy model, based on an artificial neural network, was developed to predict clinical deterioration in patients admitted in the Los Angeles hematologic malignancy unit [28]. The field of clinical medicine is undergoing a data revolution with electronic format and the growth in the collected data providing opportunities for risk prediction. In our study, we tested five algorithms, and the NB Categorical Classifier showed the best metrics for predicting refractory disease in DLBCL after a first line of treatment. With a dataset of 120 patients and no major class imbalance (33.8% observations of primary refractory disease status), we built an NBC Classifier with a ROC-AUC of 0.81 and a F1-score of 0.82 on the validation set. The NBC Classifier demonstrates strong overall predictive performance. An F1-score of 0.82 indicates that the NBC classifier has a good balance between precision and recall. An

**Table 2. Treatment characteristics and outcome.**

| | | % (n) |
|---|---|---|
| Chemotherapy (first line) | | % (124) |
| | R-CHOP | 74.2% (92) |
| | R-mini-CHOP | 13% (16) |
| | R-ACVBP | 3.1% (7) |
| | R-CEOP | 4% (5) |
| | R-MPVA | 3.2% (4) |
| PET/CT scan realization after 2 cycles of treatment | | |
| | Yes | 30% (38) |
| | No | 70% (92) |
| PET/CT scan result (n = 38) | | |
| | Complete Response | 42% (16) |
| | Partial metabolic Response | 31.5% (12) |
| | Refractory | 26.3% (10) |
| Response at first line of treatment (n = 124) | | |
| | Complete Response | 63% (78) |
| | Refractory | 33.87% (42) |
| | Not evaluated (because of dead) | 3.2% (4) |
| Relapse | Within 6 months | 1.6% (2) |
| | > 6 months | 7.2% (9) |
| Median time to relapse, months Median (min-max) | | 10 [4–36] |
| Chemotherapy (second line) n = 44 | | 84% (37) |
| | For relapse (n = 11) | 22.7% (10) |
| | For primary refractory disease (n = 42) | 77.3% (34) |
| Autograft | | 34% (15) |
| CAR-T Cell (in > = third line) | | 3.8% (5) |
| Response at the end of the study (n = 78) | | |
| | Complete Response | 89% (70) |
| | Refractory | 6.41% (5) |
| | Relapse | 2.56% (2) |
| | Not evaluated | 1.28% (1) |
| Death at the end of the study | | 40% (52) |
| Causes of death | | |
| | DLBCL | 52% (27) |
| | Septic shock | 17% (9) |
| | Sars-COV-2-19 pneumonia | 7% (4) |
| | Cardiologic reason | 7% (4) |
| | Other (other cancer, . . .) | 13.46% (7) |
| For primary refractory patients at the end of the study | | N = 42 |
| | Living | 33% (14) |
| | Complete response | 23.8% (10) |

R: Rituximab; CHOP: Rituximab Cyclophosphamide Doxorubicin Vincristine Prednisone; ACVBP: Doxorubicin Cyclophosphamide Vindesine Bleomycin Predinsone; CEOP: cyclophosphamide Etoposide Vincristine Prednisone; MPVA: Methotrexate Procarbazine Vincristine Cytarabine;

PET/CT scan: positron emission tomography/computed tomography (PET/CT) scan; DLBCL: Diffuse Large B-cell Lymphoma.

**Table 3. Ten-fold cross validation: Comparison of the performance of five machine learning models predicting primary refractory disease.**

| Model | ROC-AUC area | Accuracy | False positive rate | Sensitivity | F1-score |
|---|---|---|---|---|---|
| Naïve Bayes Categorical classifier | 0.81 | 0.83 | 10% | 0.71 | 0.82 |
| eXtremeGradientBoost | 0.74 | 0.78 | 10% | 0.57 | 0.75 |
| Random Forest classifier | 0.67 | 0.72 | 10% | 0.43 | 0.67 |
| Super Vector Machine (nuSVC) | 0.65 | 0.67 | 28% | 0.57 | 0.64 |
| Logistic Regression | 0.45 | 0.50 | 37% | 0.29 | 0.43 |

ROC-AUC: area under the curve from receiver operator characteristic.

improvement in sensitivity to capture more true positive cases will be a focus for future adjustments. This type of algorithm based on Bayes' theorem is very efficient with a low number of observations. The Naïve Bayes method combines the prior probability of an event with additional evidence to calculate a posterior probability. This method is based on applying Bayes' theorem with strong independence assumptions between the features, which is why the classifier is called "naïve". The result is the posterior probability of an outcome [29–31].

An advantage of NB Classifier is that it requires a small amount of training data to estimate the parameters necessary for classification. Another advantage is its low computational cost. Therefore, the NB Categorical Classifier is an excellent choice, especially because of the assumed independence of the predictive variables. In a study on blood transfusion in heart surgery, a NB classifier was designed to identify patients with transfusion requirements. The objective was to develop a local decision support system in this setting and a NB classifier was suitable for this purpose, as it was in our study [32].

The XGBoost model shows decent predictive performance but it is less effective in terms of discrimination than CNB Classifier, the sensitivity is relatively low, showing weakness in capturing all positive instances. The RFC model has a low false positive rate but improvements are needed to continue because sensitivity and F1-score show great difficulty in capturing all positive instances and an imbalance between precision and recall. The Support Vector Machine model and Logistic Regression demonstrate poor predictive performance with Logistic Regression having the worst results overall.

These variations in performance metrics between models highlight the differences in their underlying algorithms, complexity, and ability to capture efficiently patterns and relationships within the dataset.

The NBC classifier utilizes probabilistic principles and assumes independence between features, making it computationally efficient and robust to noisy data. On the other hand, XGBoost employs an ensemble learning technique that builds a series of decision trees sequentially, with each subsequent tree focusing on correcting the errors of the previous ones. This iterative process allows XGBoost to capture complex relationships between features in the dataset, leading to improved predictive performance. Similarly, The RF classifier constructs multiple decision trees and aggregates their predictions to make a final decision. The random selection of features and bootstrapping technique used in RF help reduce overfitting and improve generalization performance. However, its performance may vary depending on the specific characteristics of the dataset. The SVM model constructs a hyperplane or set of hyperplanes in a high-dimensional space to separate different classes. The ability of the SVM model to capture complex decision boundaries makes it suitable for tasks with non-linear relationships between features. In contrast, LR models the probability of a binary outcome based on linear combinations of predictor variables. While LR is relatively simple, it may struggle to capture complex relationships between features compared to other models like the NBC

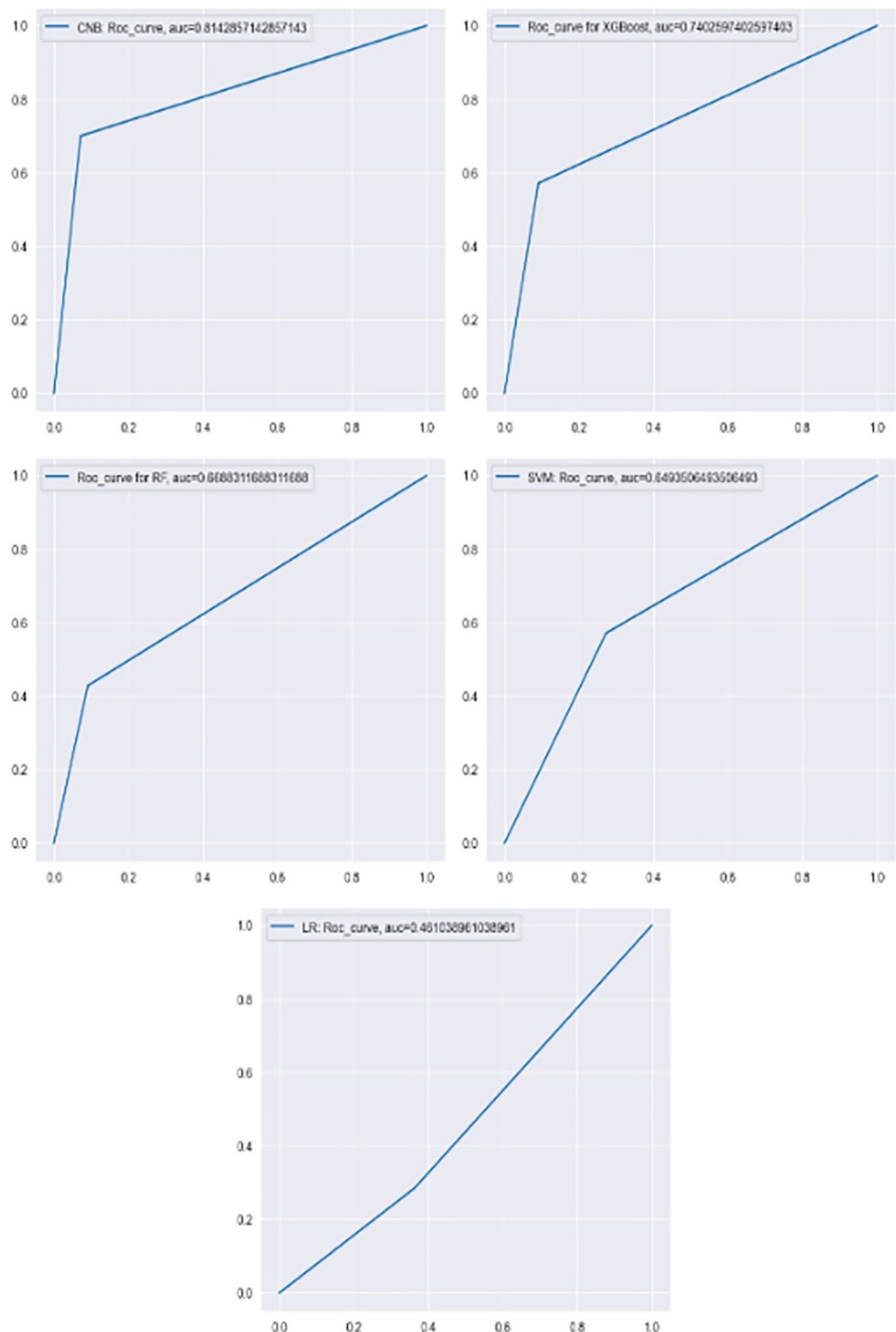

**Fig 3. The ROC-AUC curves for each model.** The characteristics of the best setting for the algorithms to which a GridSearchCV was applied: RF: max_depth: 5, n_estimators: 50, max_samples: 0.75, min_samples_splits: 10, and min_samples_leaf: 2; XGBoost: coldsample_bytree:0.8, learning_rate:0.2, max_depth:5, n_estimators:50, and subsample:0.8; and NuSVC: gamma:0.01 and kernel: rbf. All details are provided in the repository on https://github.com/MarieDetrait-MD/PrimaryRefractoryDLBCL.

**Table 4. Strengths and weaknesses of models.**

| Model | Strengths | Weaknesses |
|---|---|---|
| Naïve Bayes Categorical classifier | • ROC-AUC: 0.81, indicating a good ability to distinguish between classes<br>• Accuracy: 0.83, indicating good overall performance<br>• F1-score: 0.82, suggesting a good balance between precision and recall | • Sensitivity: 0.71, acceptable<br>• False positive rate: 10%, relatively low but still present |
| eXtremeGradientBoost | • Accuracy: 0.78, good overall performance<br>• F1-score: 0.75, showing a good balance between precision and recall | • ROC-AUC: 0.74, acceptable<br>• Sensitivity: 0.57, indicating it misses some positive cases<br>• False positive rate: 10%, relatively low but still present |
| Random Forest classifier | • Accuracy: 0.72, decent performance | • ROC-AUC: 0.67, lower than the first two models<br>• Sensitivity: 0.43, relatively low<br>• F1-score: 0.67, suggesting a poorer balance between precision and recall<br>• False positive rate: 10%, relatively low but still present |
| Super Vector Machine (nuSVC) | No significant strengths | • ROC-AUC: 0.65, among the lowest<br>• Accuracy: 0.67, lower overall performance<br>• False positive rate: 28%, the highest among the models<br>• Sensitivity: 0.57, slightly better than Random Forest |
| Logistic Regression | No significant strengths | • ROC-AUC: 0.45, the lowest, suggesting poor discrimination ability<br>• Accuracy: 0.50, close to random guessing<br>• Sensitivity: 0.29, the lowest, missing most positive cases<br>• F1-score: 0.43, the lowest, poor balance between precision and recall<br>• False positive rate: 37%, very high |

classifier. Overall, the differences in performance between the models can be attributed to their approaches to processing the data, as well as their inherent strengths and weaknesses in capturing patterns and relationships within the dataset.

The current study primarily compares models based on AUC and other common performance metrics. However, differences between these machine learning models and existing prognostic Models from the published literature have not been thoroughly explored beyond these metrics.

While the NBC classifier and other models demonstrate favorable performance according to standard metrics, ensuring proper explicability and transparency in model behavior is crucial, especially in clinical settings. Future efforts, along with the ongoing prospective study (https://clinicaltrials.gov/study/NCT06241729), will focus on a more detailed comparison with established prognostic models, particularly risk stratification based on the international Prognostic Index (IPI), cell of origin, and biomarkers such as BCL2, MYC, and BCL6, with the aim of enhancing model interpretability. This study was primarily a feasibility study and did not involve a dataset containing data unseen by the algorithms. However, in the ongoing prospective study, we have set aside a test set with unseen data for final validation to ensure the absence of overfitting. The technique known as LIME (Local Interpretable Model-agnostic Explanations) [33], which can be implemented in Python, will be employed to ensure that model decisions are clearly understood and validated by clinicians. LIME is a tool designed to explain individual predictions of machine learning models. It works by perturbing the input data around a specific instance and observing how the model's predictions change. This allows LIME to build a simpler, interpretable model that approximates the behavior of a more complex model locally. LIME helps by identifying which features, such as genetic markers or clinical history, are most influential in a prediction. This transparency is crucial in medical decision-making, as it enables clinicians to understand and trust the model's results, ultimately

leading to the final step for a well-performing model: deployment in production as a web application.

The dataset, which includes variables commonly studied in DLBCL, is a significant strength as it has the potential to be useful across a wide range of countries. Further refinement in feature engineering and selection could enhance model performance. Identifying and incorporating additional relevant features could help improve performance of the models. Indeed, increasing the number of observations and adding features to the dataset, such as biological parameters or patient and tumor characteristics, would be beneficial. However, we faced limitations due to data availability in our database. Improving data collection at our center, possibly through changes in electronic medical records, could rapidly address this limitation [34, 35]. Depending on the evolution of research and the results from other teams, new clinical, biological, and genetic parameters may be incorporated into our models in the future. For example, the polatuzumab vedotin-R-CHP modified regimen, where polatuzumab vedotin replaces vincristine, could be included as a new feature [36]. Combining multiple models through ensemble techniques, such as stacking, boosting, or bagging, could potentially provide more robust and accurate predictions. This study is a single Center retrospective study, which is another limitation. It requires validation from a large cohort to confirm the results.

In our cohort, 42 patients (33.8%) had primary refractory disease. Among these, 8 patients died due to DLBCL before salvage therapy, and 34 patients (77.3%) received salvage chemotherapy. Of the primary refractory patients, 14 (33%) were alive, and 10 (23.8%) were in remission at the end of the study. Our results are consistent with those reported in the literature [2–5]. In the study by Rovira et al. [37] on a large cohort of 468 patients, the 5-years OS was 59% and the 5-years PFS was 51% in the first line with rituximab. In our study, we similarly found a 5-year OS of 51% (95% CI, 38–61) and a 5-year PFS of 57% (95% CI, 46–67.5).

The risk factors for primary refractory disease described in the literature include older patients, low ECOG, B symptoms, elevated Ann Arbor Stage, bulky disease, extranodal involvement, bone marrow and central nervous system infiltration, elevated rate of LDH and B2 microglobulin, elevated IPI score and palliative approach [2–4, 37, 38]. In our study, we identified age (p = 0.009), Ann Arbor stage (p = 0.013), CMV infection (p = 0.012), presence of comorbidity (p = 0.019), IPI score (p< 0.001), curative intention (p<0.001), EBV infection (p = 0.008) and patient socio-economics status (p = 0.02) as significant risk factors. During data analysis, a DTC was constructed to visualize prognostic factors associated with refractory status. This model highlighted the importance of IPI score, BMI, age, time between diagnosis and treatment and the BMI category (35–40) based on feature importance.

To enhance our analysis, we examined the variables used by the three best models using the 'feature_log_prob_' attribute in the Naïve Bayes categorical classifier and feature importance for the XGBoost and Random Forest classifiers. We observed the critical importance of the IPI score and age as predictive variables, consistent with findings from previous studies [37, 38]. This analysis not only deepens our understanding but also identifies key variables that contribute to the models' performance. Our cohort is representative of the results obtained in the field, thereby supporting the development of classifier models as a proof of concept for predicting the risk of a primary refractory DLBCL.

This machine-learning approach seems particularly promising because there are currently no statistical models efficient enough to provide decision-making support to clinicians. In this study, we demonstrate that algorithms can effectively predict the refractory status of the disease using structured data from patients' medical records. Given the large number of available and effective salvage therapies, intervening quickly in the patient's therapeutic pathway appears to be the optimal and most personalized way to maximize the chances of cure while minimizing toxicity (3). Based on clinical judgment of physicians and the best algorithm

predictions, physicians can select an early treatment strategy for primary refractory DLBCL. The results of this study are encouraging; however, the next step is a prospective study (https://clinicaltrials.gov/study/NCT06241729) that aims to establish a sufficiently reliable model on previously unseen data, which allow for implementation of a decision support tool in the form of a web application.

In conclusion, we identified three interesting models for predicting refractory disease and the best was NB Categorical Classifier, achieving a ROC-AUC of 0.81 (95%CI, 0.64–0.96) and a F1-score of 0.82 on the validation set. The application of machine learning techniques can significantly contribute to the management of DLBCL patients. These algorithms hold the potential to assist clinicians in making informed decisions regarding treatment strategies, enabling the personalization of therapies based on individual patient data. To validate these findings on a broader scale, a prospective multicenter study could confirm the results in a larger cohort and demonstrate the value of this technology in the complex management of primary refractory disease in DLBCL patients.

## Author Contributions

**Conceptualization:** Marie Y. Detrait.

**Data curation:** Stéphanie Warnon, Stéphanie De Prophétis.

**Formal analysis:** Marie Y. Detrait, Stéphanie Warnon.

**Methodology:** Marie Y. Detrait, Delphine Pranger.

**Project administration:** Stéphanie Warnon, Stéphanie De Prophétis, Amandine Hansenne, Juliette Raedemaeker, Valérie Robin, Géraldine Verstraete, Aline Gillain, Nicolas Depasse, Pierre Jacmin.

**Resources:** Marie Y. Detrait, Stéphanie De Prophétis.

**Supervision:** Delphine Pranger.

**Validation:** Marie Y. Detrait.

**Visualization:** Marie Y. Detrait, Stéphanie Warnon.

**Writing – original draft:** Marie Y. Detrait.

**Writing – review & editing:** Stéphanie Warnon, Raphaël Lagasse, Laurent Dumont, Aline Gillain, Delphine Pranger.

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
