## [Decision Letter · Decision Letter 0]

16 May 2024

PONE-D-24-07639A Machine learning approach in a monocentric cohort for predicting primary refractory disease in Diffuse Large B-Cell Lymphoma patientsPLOS ONE

Dear Dr. Detrait,

Thank you for submitting your manuscript to PLOS ONE. After careful consideration, we feel that it has merit but does not fully meet PLOS ONE’s publication criteria as it currently stands. Therefore, we invite you to submit a revised version of the manuscript that addresses the points raised during the review process.

We look forward to receiving your revised manuscript.

Kind regards,

Andres Mauricio Acevedo-Melo, M.D.

Academic Editor

PLOS ONE

Journal Requirements:

3. In the online submission form, you indicated that "Data are available from the Ethics Committee (contact via Dr Marie Detrait) for researchers who meet the criteria for access to confidential data."

4. Please amend the manuscript submission data (via Edit Submission) to include author Dr. Valérie Robin.

**Additional Editor Comments:**

In this article, the authors present an interesting exploratory study on prognosis assisted by machine learning (ML) in a monocentric study of 130 adults diagnosed with Diffuse Large B Cell Lymphoma (DLBCL) in Belgium. By comparing the discriminatory capabilities of five different supervised ML models with relevant patient and disease characteristics, they have demonstrated the feasibility of supporting primary refractory disease prognostication in an offline validation setting, which can be further validated in prospective cohorts. As an original investigation into AI-assisted decision-making processes in clinical settings for a high-burden disabling disease, this manuscript is within the scope of PLOS ONE and meets the publication criteria. However, some issues need to be addressed before the manuscript can be considered for full-text publication. Please find reviewers comments and provide point-by-point response. Additionally, adherence to artificial intelligence-assisted prognosis reporting guidelines is recommended (TRIPOD-AI checklist attached). Please find the main points that need to be addressed:

• Provide a comprehensive explanation and reporting of model performance measures (calibration and discrimination) in both training and validation datasets for all five ML models to allow for a thorough critical evaluation.

• Expand the discussion to consider existing evidence, study strengths and limitations, and potential barriers to further implementation in clinical settings.

• Organize the manuscript coherently and conduct a thorough grammar and style review to correct typos in the manuscript (e.g., "septic choc" and "TEP-Scanner").

Reviewers' comments:

Reviewer's Responses to Questions

**Comments to the Author**

1. Is the manuscript technically sound, and do the data support the conclusions?

Reviewer #1: Partly

Reviewer #2: Partly

2. Has the statistical analysis been performed appropriately and rigorously? 

Reviewer #1: Yes

Reviewer #2: Yes

3. Have the authors made all data underlying the findings in their manuscript fully available?

Reviewer #1: Yes

Reviewer #2: Yes

4. Is the manuscript presented in an intelligible fashion and written in standard English?

Reviewer #1: No

Reviewer #2: Yes

5. Review Comments to the Author

Reviewer #1: The manuscript presents a machine learning approach in a monocentric cohort for predicting primary refractory disease in diffuse large B-cell lymphoma (DLBCL) patients. To enhance the paper's quality, several minor revisions are necessary to address the following concerns:

1.The manuscript requires language refinement to correct grammatical errors and improve overall readability.

2.It would be beneficial for the authors to provide a flowchart outlining the structure of the paper to help readers gain a comprehensive overview.

3.The authors should offer a more detailed discussion on the advantages and disadvantages of the five machine learning models employed in the study.

4.Since the analysis includes ROC curves, it is advisable for the authors to plot these curves for clarity.

5.Before introducing the methodology, the authors should cite literature on machine learning for predicting DLBCL to demonstrate the feasibility of their approach.

6.A comprehensive discussion on the strengths and limitations of the study is warranted, including any potential areas for enhancement.

7.The current literature lacks recent citations from the past three years. Therefore, it is essential to incorporate more up-to-date references to enrich the study.

Reviewer #2: This manuscript by Detrait, et al appears to be an important advance and application in the field of computational science and clinic, since they are incorporating a machine learning model including clinical features to predict primary refractory DLBCL.

Overall I am enthusiastic about this manuscript as a brief initial description of a potentially useful model for the field in particular for countries and regions where there is no access to molecular or advance techniques to predict outcomes. The comments below address areas to strengthen or clarify the findings in the manuscript:

1. It would be better if the authors include the ROC curve figures to have an illustration that allows visually compare the performance among different models

2. When the author describe the univariate analysis, I think it preferred to show the p-values as recommended:"There are two ways to report p values: (a) report p value based on the α level determined, e.g., “p > 0.05 or p < 0.05” or “p > 0.01 or p < 0.01” and (b) report the exact p value (the posteriori probability reported by the statistical software). If the exact p value is less than 0.001, it is conventional to state merely p < 0.001"

3. What is the F-score described in the univariate analysis ? is it the same as F1-Score? it is not clear to me, as it is described in integers for the different features included in the model and not the F1-Score for the model itself. Please clarify

4. F1-score should be included in table 3.

5. I understand that you share the code only in specific cases, trying to preserve confidentiality for patient data. However, I believe that if you want your model to be validate by other groups, a website/blog with a tool where a clinician might introduce data an have a prediction using your model would be ideal. For doing that you don't have to share patient data. This is a suggestion that the authors may consider, but it is not compelled to the authors to fulfill.

6. PLOS authors have the option to publish the peer review history of their article (what does this mean?). If published, this will include your full peer review and any attached files.

Reviewer #1: **Yes: **Yidong Zhu

Reviewer #2: **Yes: **Bonell Patiño-Escobar

---

## [Author Response · Author response to Decision Letter 0]

1 Jul 2024

To Editorial Board of PlosOne

Dear Editor, 

Subject : Response to Reviewers’Comments on Manuscript ID [PONE-D-24-07639]

We sincerely thank you for the valuable feedback you provided on our manuscript titled « [A Machine learning approach in a monocentric cohort for predicting primary refractory disease in Diffuse Large B-Cell Lymphoma patients] ». We have carefully considered each comment and have made the necessary revisions to address them. Below, we provide detailed responses to each reviewers’s comments and yours. 

• Provide a comprehensive explanation and reporting of model performance measures (calibration and discrimination) in both training and validation datasets for all five ML models to allow for a thorough critical evaluation.

In this study, the models were evaluated using the validation set. This study was primarily a feasability study, and there was no dataset with data unseen by the algorithms. In the ongoing prospective study, a test set containing data unseen by the algorithms has been reserved for the purpose of final validation and potential deployment into production, provided the results are satisfactory. The article, metrics, and models have been revised in light of the recommendations (TRIPOD-AI). We have expanded the discussion and included a comprehensive report detailing the strengths and weaknesses of the models (Table 4) to better reflect their perfomance. 

Otherwise, we have set up a repository on Github with the dataset and the informatics code at https://github.com/MarieDetrait-MD/PrimaryRefractoryDLBCL. 

• Expand the discussion to consider existing evidence, study strengths and limitations, and potential barriers to further implementation in clinical settings.

The discussion has been expanded to address these questions and requests.

• Organize the manuscript coherently and conduct a thorough grammar and style review to correct typos in the manuscript (e.g., "septic choc" and "TEP-Scanner").

The article has been revised accordingly. 

Reviewer #1: 

1.The manuscript requires language refinement to correct grammatical errors and improve overall readability. 

The manuscript has undergone a thorough language refinement process to correct grammatical errors enhance overall readability. 

2.It would be beneficial for the authors to provide a flowchart outlining the structure of the paper to help readers gain a comprehensive overview.

A comprehensive flowchart outlining the structure of the paper has been included to provide readers with a clear overview. This flowchart represents the key components and flow of the study.

3.The authors should offer a more detailed discussion on the advantages and disadvantages of the five machine learning models employed in the study.

We have provided an expanded discussion on the advantages and disadvantages of the five machine learning models employed in the study. 

4.Since the analysis includes ROC curves, it is advisable for the authors to plot these curves for clarity.

ROC curves for the five machine learning models have been plotted and included in the manuscript. These plots are presented in the ‘Results‘ section, providing a visual representation of the models’ performance aiding in the comparison of their effectiveness.

5.Before introducing the methodology, the authors should cite literature on machine learning for predicting DLBCL to demonstrate the feasibility of their approach.

The literature review has been expanded and recent references have been added for machine learning research in prediction, prognosis cancer and diffuse large B cell lymphoma. The literature review demonstrates the feasability of our approach and positions our study within the context of existing research in diffuse large B cell lymphoma and machine learning. These citations are included in the ‘Introduction’ section.

References : 

8.Gaur K, Jagtap MM. Role of Artificial Intelligence and Machine Learning in Prediction, Diagnosis, and Prognosis of Cancer. (2022) Cureus. 14(11):31008.

9. Zhang B, Shi H, Wang H. Machine Learning and AI in Cancer Prognosis, Prediction, and Treatment Selection: A Critical Approach. (2023) J Multidiscip Healthc. 16:1779-1791.

12.Steen CB, Luca BA, Esfahani MS, Azizi A, Sworder BJ, et al. (2021) The landscape of tumor cell states and ecosystems in diffuse large B cell lymphoma. Cancer Cell. 39(10):1422-1437.

13. Zhu Y, Ning Z, Li X, Lin Z. Machine learning algorithms identify target genes and the molecular mechanism of matrine against diffuse large B-cell lymphoma. (2024) Curr Comput Aided Drug Des. 20(6):847-859.

6.A comprehensive discussion on the strengths and limitations of the study is warranted, including any potential areas for enhancement.

The discussion has been reviewed to adress this point. 

7.The current literature lacks recent citations from the past three years. Therefore, it is essential to incorporate more up-to-date references to enrich the study.

The manuscript has been updated to include recent citations from the past three years. These up-to-date references enrich the study by incorporating the latest findings and advancements in the field. These citations are included in the ‘introduction’ section. 

References : 

8.Gaur K, Jagtap MM. Role of Artificial Intelligence and Machine Learning in Prediction, Diagnosis, and Prognosis of Cancer. (2022) Cureus. 14(11):31008.

9. Zhang B, Shi H, Wang H. Machine Learning and AI in Cancer Prognosis, Prediction, and Treatment Selection: A Critical Approach. (2023) J Multidiscip Healthc. 16:1779-1791.

12.Steen CB, Luca BA, Esfahani MS, Azizi A, Sworder BJ, et al. (2021) The landscape of tumor cell states and ecosystems in diffuse large B cell lymphoma. Cancer Cell. 39(10):1422-1437.

13. Zhu Y, Ning Z, Li X, Lin Z. Machine learning algorithms identify target genes and the molecular mechanism of matrine against diffuse large B-cell lymphoma. (2024) Curr Comput Aided Drug Des. 20(6):847-859.

Reviewer #2 : 

1. It would be better if the authors include the ROC curve figures to have an illustration that allows visually compare the performance among different models. 

The Roc-AUC figure for for the five machine learning models have been plotted and included in the manuscript. These plots are presented in the ‘Results ‘ section, providing a visual representation of the models’ performance aiding in the comparison of their effectiveness.

2. When the author describe the univariate analysis, I think it preferred to show the p-values as recommended:"There are two ways to report p values: (a) report p value based on the α level determined, e.g., “p > 0.05 or p < 0.05” or “p > 0.01 or p < 0.01” and (b) report the exact p value (the posteriori probability reported by the statistical software). If the exact p value is less than 0.001, it is conventional to state merely p < 0.001"

The p value has been corrected. 

3. What is the F-score described in the univariate analysis ? is it the same as F1-Score? it is not clear to me, as it is described in integers for the different features included in the model and not the F1-Score for the model itself. Please clarify

The F-score was Feature-importance. The F1-score for each model has been added and discussed. A reference has been added to characterize the different measures. 

24 Sokolova M, Lapalme G (2009) A systematic analysis of performance measures for classification tasks. Inf. Process. Manage. 45(4):427-437. 

4. F1-score should be included in table 3. 

F1-score has been included. 

5. I understand that you share the code only in specific cases, trying to preserve confidentiality for patient data. However, I believe that if you want your model to be validate by other groups, a website/blog with a tool where a clinician might introduce data an have a prediction using your model would be ideal. For doing that you don't have to share patient data. This is a suggestion that the authors may consider, but it is not compelled to the authors to fulfill. 

We have set up a repository on Github with the dataset and the informatics code at https://github.com/MarieDetrait-MD/PrimaryRefractoryDLBCL. 

In conclusion, we believe that the revisions made have significantly improved the manuscript’s quality and clarity. We appreciate the opportunity to address the reviewers’s comments and thank the editorial board for their consideration of our revised manuscript. 

Please find attached the revised manuscript, which incorporates all suggested changes. We look forward to your favorable consideration of our manuscript for publication in PlosOne. 

Sincerely, 

Dr Marie DETRAIT

---

## [Decision Letter · Decision Letter 1]

6 Aug 2024

PONE-D-24-07639R1A Machine learning approach in a monocentric cohort for predicting primary refractory disease in Diffuse Large B-Cell Lymphoma patientsPLOS ONE

Dear Dr. Detrait,

Thank you for submitting your manuscript to PLOS ONE. After careful consideration, we feel that it has merit but does not fully meet PLOS ONE’s publication criteria as it currently stands. Therefore, we invite you to submit a revised version of the manuscript that addresses the points raised during the review process.

We look forward to receiving your revised manuscript.

Kind regards,

Andres Mauricio Acevedo-Melo, M.D.

Academic Editor

PLOS ONE

Journal Requirements:

Additional Editor Comments:

The manuscript presents an exploratory study on prognosis assisted by machine learning (ML) in a monocentric study of 130 adults diagnosed with Diffuse Large B Cell Lymphoma (DLBCL) in Belgium. The study aims to evaluate five supervised machine learning models for predicting primary refractory DLBCL. By comparing the discriminatory capabilities of these ML models with relevant patient and disease characteristics, the authors explore their feasibility for further validation in prospective cohorts. This study explores a compelling application of artificial intelligence in disease prognostication, making it highly relevant to the scope of PLOS ONE. The innovative approach of using ML for predicting DLBCL outcomes could significantly contribute to advancements in the field.

Specific Areas for Improvement Before Publication

While the authors have submitted a revised version that two expert reviewers have deemed ready for publication, I have noticed several issues in the revised manuscript that need to be addressed:

1. The authors have changed the model performance measures from the original submission to the first revision (R1) without providing an adequate explanation.

2. There is no clear comparison of ML models to existing published prognostication models beyond AUC. While Logistic regression appears to underperform, models seem to overestimate their performance without proper explicability. This is recommended per original TRIPOD statement and update TRIPOD-AI. Please see: BMJ 2024; 385 doi: https://doi.org/10.1136/bmj-2023-078378

3. An additional author has been added to the original list without a proper explanation.

Reviewers' comments:

Reviewer's Responses to Questions

**Comments to the Author**

1. If the authors have adequately addressed your comments raised in a previous round of review and you feel that this manuscript is now acceptable for publication, you may indicate that here to bypass the “Comments to the Author” section, enter your conflict of interest statement in the “Confidential to Editor” section, and submit your "Accept" recommendation.

Reviewer #1: All comments have been addressed

Reviewer #2: All comments have been addressed

2. Is the manuscript technically sound, and do the data support the conclusions?

Reviewer #1: Yes

Reviewer #2: Yes

3. Has the statistical analysis been performed appropriately and rigorously? 

Reviewer #1: Yes

Reviewer #2: Yes

4. Have the authors made all data underlying the findings in their manuscript fully available?

Reviewer #1: Yes

Reviewer #2: Yes

5. Is the manuscript presented in an intelligible fashion and written in standard English?

Reviewer #1: Yes

Reviewer #2: Yes

6. Review Comments to the Author

Reviewer #1: The authors have satisfactorily addressed my concerns regarding the manuscript. The revised version is suitable for publication.

Reviewer #2: All the observations have been addressed carefully and the authors have agreed that the observations might improve clarity for the paper

7. PLOS authors have the option to publish the peer review history of their article (what does this mean?). If published, this will include your full peer review and any attached files.

Reviewer #1: **Yes: **Yidong Zhu

Reviewer #2: **Yes: **Bonell Patino-Escobar

---

## [Author Response · Author response to Decision Letter 1]

27 Aug 2024

August 26, 2024

To Editorial Board of PlosOne

Dear Editor, 

Subject: Response to Editor’s Comments on Manuscript ID [PONE-D-24-07639]

We would like to express our sincere gratitude for the insightful feedback provided on our manuscript titled « [A Machine learning approach in a monocentric cohort for predicting primary refractory disease in Diffuse Large B-Cell Lymphoma patients] ». We have carefully reviewed each comment and have made the necessary revisions to address them comprehensively. 

Below, we provide detailed responses to the specific points raised:

1. The authors have changed the model performance measures from the original submission to the first revision (R1) without providing an adequate explanation.

We apologize for not providing a clear explanation initially. The modification was due to a change in the data preparation process. Originally, age was discretized, but when preparing the revised version, we decided to retain the original continuous age variable, as age categories were already pre-defined (age category one (18-40), age category two (40-60), age category three (60-80), age category four (>80)). This change, indeed, had an impact on the models’ performance metrics. We appreciate your attention to this detail, and we should have communicated this change in our first rebuttal letter. 

2. There is no clear comparison of ML models to existing published prognostication models beyond AUC. While Logistic regression appears to underperform, models seem to overestimate their performance without proper explicability. This is recommended per original TRIPOD statement and update TRIPOD-AI. Please see: BMJ 2024; 385 doi: https://doi.org/10.1136/bmj-2023-078378

a. To enhance explainability of our models, we analyzed the most important features using the ‘feature-importance’ attribute for both the XGBoost and Random Forest models. Additionally, we examined the log-likelihoods of conditional probabilities across different classes and variables using the ‘feature_log_prob_’ attribute in the Naive Bayes classifier. This analysis helped us identify the variables that contributed most significantly to the model’s predictions. We then compared these variables with established prognostic indicators in the literature. We found that the variables utilized by the Naive Bayes classifier were particularly relevant for accurate predictions. Moreover, the IPI score and age were consistently identified as significant across all three models. These findings have now been integrated into the manuscript and are discussed in detail. 

b. We recognize that our comparison of models in the current study primarily focused on AUC and other common performance metrics. A comprehensive comparison with existing prognostic models from the literature, beyond these metrics, was not fully explored. Although the Naive Bayes classifier and other models demonstrated promising results based on standard metrics, we acknowledge the importance of ensuring explainability and transparency of model behavior, particularly in clinical contexts. Moving forward, including in our ongoing prospective study, we plan to conduct a more thorough comparison with established prognostic models, particularly focusing on the IPI score, cell of origin, and biomarkers such as BCL2, MYC, and BCL6. We also intend to enhance the interpretability of our models. Specifically, we will employ the LIME (Local Interpretable Model-agnostic Explanations) technique, which will be implemented in Python for our prospective study, to ensure that model decisions are transparent and can be endorsed by clinicians. 

It is important to note that this study was primarily a feasibility study and did not involve a dataset containing data unseen by the algorithms. However, in the ongoing prospective study, we have set aside a test set with unseen data for final validation to ensure the absence of overfitting. 

We appreciate the reference to the “TRIPOD+AI statement : updated guidance for reporting clinical prediction models that use regression or machine learning methods“ by Gary S collins et al., which provides invaluable guidance for our ongoing and future studies. Thank you for highlighting this crucial aspect, which we agree warrants discussion in our manuscript. 

We have also added a reference for the LIME technique as follows: 

33. Ribeiro MT, Singh S, Guestrin C (2016) “Why should I trust you?”: explaining the predictions of any classifier. Proceedings of the 22nd ACM SIGKDD International Conference on Knowledge Discovery and Data Mining. New York: ACM, 1135-1144. 

3. An additional author has been added to the original list without a proper explanation.

You are correct in noting this discrepancy. Dr. V. Robin was inadvertently omitted when the authors’ names were initially submitted online; although she was correctly listed as an author in the manuscript itself. This oversight has been corrected, and we appreciate your diligence in bringing this to our attention. 

In conclusion, we believe that the revisions made have significantly improved the quality of our manuscript. We appreciate the opportunity to address these important comments and thank you and the editorial board for your thoughtful consideration of our revised manuscript. Please find attached the revised manuscript, which incorporates all suggested changes. We look forward to your favorable consideration of our manuscript for publication in PlosOne. 

Sincerely, 

Dr Marie DETRAIT

---

## [Decision Letter · Decision Letter 2]

17 Sep 2024

A Machine learning approach in a monocentric cohort for predicting primary refractory disease in Diffuse Large B-Cell Lymphoma patients

PONE-D-24-07639R2

Dear Dr. Detrait,

We’re pleased to inform you that your manuscript has been judged scientifically suitable for publication and will be formally accepted for publication once it meets all outstanding technical requirements.

Kind regards,

Jian Wu, M.D, Ph.D

Academic Editor

PLOS ONE

Additional Editor Comments (optional):

Reviewers' comments:

Reviewer's Responses to Questions

**Comments to the Author**

1. If the authors have adequately addressed your comments raised in a previous round of review and you feel that this manuscript is now acceptable for publication, you may indicate that here to bypass the “Comments to the Author” section, enter your conflict of interest statement in the “Confidential to Editor” section, and submit your "Accept" recommendation.

Reviewer #1: All comments have been addressed

2. Is the manuscript technically sound, and do the data support the conclusions?

Reviewer #1: Yes

3. Has the statistical analysis been performed appropriately and rigorously? 

Reviewer #1: Yes

4. Have the authors made all data underlying the findings in their manuscript fully available?

Reviewer #1: Yes

5. Is the manuscript presented in an intelligible fashion and written in standard English?

Reviewer #1: Yes

6. Review Comments to the Author

Reviewer #1: (No Response)

7. PLOS authors have the option to publish the peer review history of their article (what does this mean?). If published, this will include your full peer review and any attached files.

Reviewer #1: **Yes: **Yidong Zhu

---

## [Editor Report · Acceptance letter]

20 Sep 2024

PONE-D-24-07639R2 

PLOS ONE

Dear Dr. Detrait, 

I'm pleased to inform you that your manuscript has been deemed suitable for publication in PLOS ONE. Congratulations! Your manuscript is now being handed over to our production team.

Kind regards, 

on behalf of

Dr. Jian Wu 

Academic Editor

PLOS ONE